

# Anthropogenic climate change versus internal climate variability: impacts on snow cover in the Swiss Alps

**Fabian Willibald**[1,2]**, Sven Kotlarski**[3]**, Adrienne Grêt-Regamey**[1,2]**, and Ralf Ludwig**[4]

[1]Planning of Landscape and Urban Systems, Institute for Spatial and Landscape Planning,
ETH Zurich, Zurich, Switzerland
[2]TS1 Institute of Science, Technology and Policy, ETH Zurich, Zurich, Switzerland
[3]Federal Office of Meteorology and Climatology MeteoSwiss, Zurich-Airport, Switzerland
[4]Department of Geography, Ludwig-Maximilians-University Munich, Munich, Germany

**Correspondence:** Fabian Willibald (fabian.willibald@istp.ethz.ch)

**Abstract.** TS2 Snow is a sensitive component of the climate system. In many parts of the world, water stored as snow is a vital resource for agriculture, tourism and the energy sector. As uncertainties in climate change assessments are still relatively large, it is important to investigate the interdependencies between internal climate variability and anthropogenic climate change and their impacts on snow cover. We use regional climate model data from a new single-model large ensemble with 50 members (ClimEX LE) as a driver for the physically based snow model SNOWPACK at eight locations across the Swiss Alps. We estimate the contribution of internal climate variability to uncertainties in future snow trends by applying a Mann–Kendall test for consecutive future periods of different lengths (between 30 and 100 years) until the end of the 21st century. Under RCP8.5, we find probabilities between 15 % and 50 % that there will be no significant negative trend in future mean snow depths over a period of 50 years. While it is important to understand the contribution of internal climate variability to uncertainties in future snow trends, it is likely that the variability of snow depth itself changes with anthropogenic forcing. We find that relative to the mean, interannual variability of snow increases in the future. A decrease in future mean snow depths, superimposed by increases in interannual variability, will exacerbate the already existing uncertainties that snow-dependent economies will have to face in the future.

# 1 Introduction

In large parts of the world, water stored in snow is a vital resource for water management with regard to agriculture and power generation. Snow cover extent and duration are also premises for winter tourism. As part of the climate system, snow influences the energy balance and heat exchange and is therefore a crucial component for land surface–atmosphere interactions (Hadley and Kirchstetter, 2012; Henderson et al., 2018). At the same time, snow is very sensitive to changes in the climate system. Several studies have analyzed trends in historical snow cover, but there is not a uniform pattern across the world. While there are many regions where snow cover and depth are decreasing, there are also areas that show no trend or even increasing snow depths (Dyer and Mote, 2006; Schöner et al., 2019; Zhang and Ma, 2018). These contrasting findings can be attributed to spatial and temporal climate variability, from global to local scales.

In addition to studies dealing with historical snow trends, many studies investigate the potential impacts of anthropogenic climate change on snowpack. The vast majority of those studies conclude that anthropogenic climate change will significantly reduce snow cover. In a global analysis, Barnett et al. (2005) find that reduced snow cover will lead to severe consequences for future water availability. On the continental scale, Brown and Mote (2009) simulate a serious decrease in seasonal snow cover in a future climate. On the regional scale, Marty et al. (2017) and Verfaillie et al. (2018) compared the impact of different emission scenarios on fu-

**Published by Copernicus Publications on behalf of the European Geosciences Union.**

ture snowpack in the Swiss and French Alps, respectively, and found a significant reduction under all scenarios and for all elevation zones. Ishida et al. (2019) and Khadka et al. (2014) found that climate change will lead to severe shifts in snow regimes in California and Nepal, respectively.

However, the potential impacts of climate change on snow hydrology remain disputed, largely because of uncertainties attributed to future greenhouse gas emissions, model uncertainties and internal climate variability (ICV; Beniston et al., 2018). ICV is defined as the natural fluctuations in the climate system that arise in the absence of any radiative forcing (Hawkins and Sutton, 2009). Typically, studies compare different emission scenarios to tackle the uncertainties related to future greenhouse gas emissions and use a multimodel ensemble approach to estimate uncertainties related to model uncertainties (Frei et al., 2018; Marty et al., 2017). While future greenhouse gas emissions and model uncertainties are the subject of multiple studies (Kudo et al., 2017), only very few studies investigate the impact of ICV on snow (Fyfe et al., 2017).

When using a multimodel ensemble approach, it is difficult to quantify ICV impacts or separate contributions from ICV and external forcing since it is very challenging to distinguish between model uncertainties and ICV. The reason for this is that intermodel spread is commonly derived from the complex coupling of different model structures, parameterizations and atmospheric initial conditions (Gu et al., 2019). Nevertheless, a few studies estimated the fraction of uncertainty in the hydrometeorological process chain ranging from different emission scenarios to the applied impact model and found that on shorter timescales, ICV represents the single most important source of uncertainty (Fatichi et al., 2014; Lafaysse et al., 2014). To investigate the combined influences of ICV and anthropogenic forcing (atmospheric concentration of greenhouse gases and aerosols), single-model large ensembles, generated by small differences in the models' initial conditions, have been developed (Deser et al., 2012; Kay et al., 2015; Leduc et al., 2019). Those studies allow a probabilistic assessment of ICV. Deser et al. (2012), for example, used a 40-member initial-condition ensemble to estimate the contribution of ICV in future North American climate, and Fischer et al. (2013) used a 21-member single-model ensemble to assess the role of ICV in future climate extremes. Mankin and Diffenbaugh (2014) investigated the influence of precipitation variability on near-term Northern Hemisphere snow trends. Because of high computational costs, those ensembles are usually used on the scale of general circulation models; dynamically downscaled single-model large ensembles, using a regional climate model (RCM), are very rare. To our knowledge, only Fyfe et al. (2017) used snow water equivalent from a downscaled single-model large ensemble to estimate the impact of ICV on near-term snowpack loss over the United States. Nevertheless, the combined effects of ICV and external forcing on snow remain insufficiently quantified, and a single-model large ensemble has not yet been used to drive a snowpack model for regional impact studies.

While it is important to estimate the contribution of ICV to uncertainties in future snow trends, it is just as important to investigate the interannual variability (IAV) of snow itself, which is defined as the year-to-year deviation from a long-term mean (He and Li, 2018) and which is likely to change under a future climate (IPCC, 2013). While the response of IAV of snow depths to anthropogenic climate change can pose risks and increasing uncertainties for agriculture, power generation and winter tourism, these processes are only inadequately studied. Again, by complementing multimodel-based approaches and by separating ICV from forced responses, a single-model large ensemble can help answer how interannual variability might respond to changes in climatic forcing.

We state the following hypotheses and aim to answering two research questions: first, ICV is a major source of uncertainty in trends of future Alpine snow depth. Our research question is as follows: what are the uncertainties in future trends in Alpine snow depth attributed to ICV? Second, IAV of snow depth will change with anthropogenic climate forcing. Hence, our research question is as follows: how does IAV of snow depth change with anthropogenic climate forcing?

To answer these questions, we use a dynamically downscaled single-model large ensemble to drive a state-of-the-art, physically based snowpack model for eight stations across the Swiss Alps. In the first part, we assess the ensemble mean change of snow depths in a future climate. In the second part, we assess the probabilities for a significant reduction in annual mean and maximum snow depth in the presence of ICV. In the third part, we quantify how interannual variability of snow depth might change in a future climate.

## 2    Methods and data

### 2.1    Case studies

This study assesses the interdependencies between ICV and anthropogenic climate change for eight stations across the Swiss Alps. The choice of these case studies was driven by the availability of long-term observations needed for model validation and bias correction. The selected stations are considered representative as they spread over the whole ridge of the Swiss Alps and cover the northern and southern parts of the mountain range as well as cover elevations between 1060 and 2540 m a.s.l. (Fig. 1, Table 1). Observational data of temperature, precipitation, wind speed, humidity and incoming shortwave radiation were provided by the Swiss Federal Office of Meteorology and Climatology (MeteoSwiss: https://www.meteoswiss.admin. ch/home/services-and-publications/beratung-und-service/

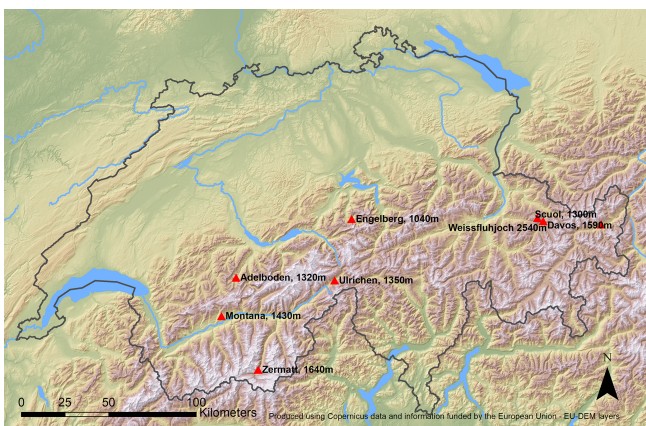

**Figure 1.** Overview map of case studies used for this study (produced using Copernicus data and information funded by the European Union EU-DEM CE2 layers).

**Table 1.** Summary of case studies.

| Station | ID | Coordinates (lat °N, long °E) | Elevation (m a.s.l.) |
|---|---|---|---|
| Adelboden | ABO | 46.5, 7.6 | 1325 |
| Engelberg | ENG | 46.8, 8.4 | 1060 |
| Davos | DAV | 46.8, 9.9 | 1560 |
| Montana | MON | 46.3, 7.5 | 1590 |
| Scuol | SCU | 46.8, 10.3 | 1298 |
| Ulrichen | ULR | 46.5, 8.3 | 1366 |
| Weissfluhjoch | WFJ | 46.8, 9.8 | 2540 |
| Zermatt | ZER | 46.0, 7.8 | 1600 |

datenportal-fuer-lehre-und-forschung.html, last access: TS3 ) and the WSL CE1 Institute of Snow and Avalanche Research (SLF) in a 3-hourly temporal resolution. Daily measurements of snow depth used for model validation stem from the same sources. The temporal coverage of the observational data for the purpose of bias correction and model validation ranges from 1983 to 2010.

## 2.2 The SNOWPACK model

SNOWPACK is a physically based, one-dimensional snow cover model (Lehning et al., 1999). It was originally developed for avalanche forecasting (Lehning et al., 2002a) but is increasingly used for climate change studies (Katsuyama et al., 2017; Schmucki et al., 2015). SNOWPACK is highly advanced with regard to snow microstructural detail. The model uses a Lagrangian finite-element method to solve the partial differential equations regulating the mass, energy and momentum transport within the snowpack. Calculations of the energy balance, mass balance, phase changes, water movement and snow transport by wind are included in the model. Finite elements can be added through solid precipitation and subtracted by erosion, melt water runoff, evapora-

tion or sublimation (Lehning et al., 1999, 2002a, b; Schmucki et al., 2014). Recommended temporal resolutions range from 15 min (e.g., for avalanche forecasting) to 3 h (e.g., for long-term climate studies). Minimum meteorological input for SNOWPACK is air temperature, relative humidity, incoming longwave radiation, wind speed and precipitation. Due to the lack of measurements, incoming longwave radiation had to be estimated based on air temperature, incoming shortwave radiation and relative humidity; CE3 using the parameterization by Konzelmann et al. (1994), Shakoor et al. (2018) and Shakoor and Ejaz (2019) applied this method for multiple sites and elevations and found that it gives reliable estimates. As wind-induced gauge undercatch underestimates precipitation, especially for mixed and solid precipitation, we do not use the original measured precipitation data to run the model. As described in Schmucki et al. (2014), precipitation was undercatch-corrected by applying a method developed by Hamon (1973), using a function of wind speed and temperature.

Soil layers are not included in our model setup. Therefore, ground surface temperature is determined as a Dirichlet boundary condition (Schmucki et al., 2014) and soil temperature is fixed at $0\,°C$. To account for site-specific characteristics, we calibrated roughness length and rainfall–snowfall threshold temperature. For roughness length, we used values between 0.01 and 0.08. For rainfall–snowfall discrimination, we used threshold temperatures between 0.2 and $1.2\,°C$, which lies well within the calibration ranges between $-0.4$ and $2.4\,°C$ based on results from Jennings et al. (2018). The calibration was carried out individually for each site. A threshold of $50\,\%$ in relative humidity was set for all stations for rainfall–snowfall discrimination.

SNOWPACK usually operates on very high temporal resolutions. After an initial sensitivity analysis, to get better simulation results, the meteorological input was resampled within SNOWPACK to a resolution of 1 h. Precipitation was evenly disaggregated from a 3-hourly time step to 1 h, while the remaining parameters were linearly interpolated. The case studies were validated based on observed daily and monthly snow depths.

## 2.3 Simulation data

In this study, we use climate model data from a new single-model large ensemble, hereafter referred to as the ClimEx large ensemble (ClimEx LE; Leduc et al., 2019; von Trentini et al., 2019), to analyze the combined influence of ICV and climate change on future snow depth. The ClimEx LE consists of 50 members of the Canadian Earth System Model (CanESM2; Arora et al., 2011), which is downscaled for a European and North American domain by the Canadian Regional CE4 Model (CRCM5; Šeparović et al., 2013). Each ensemble member undergoes the same external forcing and starts with identical initial conditions in the ocean, land and sea-ice model components but slightly different initial

conditions in the atmospheric model. The 50 members of the CanESM2 originate from five families of simulations, each starting at different 50-year intervals of a preindustrial run with a stationary climate and ranging from 1850 to 1950. In 1950, small differences in the initial conditions are used to separate each family into 10 members. After applying small atmospheric perturbations in the initial conditions, each member evolves chaotically over time. Therefore, the model spread shows how much the climate can vary as a result of random internal variations (Deser et al., 2012). This makes all 50 members equally likely, plausible realizations of climate change over the next century. Until 2005 the ensemble is driven by observed forcing; from 2006 to 2099 all simulations are forced with concentrations according to the RCP8.5 scenario (Moss et al., 2010).

The 50 members are then dynamically downscaled over Europe using CRCM5 with a horizontal grid size of 0.11° on a rotated latitude–longitude grid, corresponding to a 12 km resolution (von Trentini et al., 2019). Detailed information on the design of the experiment can be found in Leduc et al. (2019). The ClimEx LE provides meteorological data at a 3-hourly temporal resolution.

In an intercomparison experiment by von Trentini et al. (2019), the ClimEx LE was compared to 22 members of the EURO-CORDEX multimodel ensemble. It was found that the ClimEx LE shows stronger climate change signals, and the single-model spread is usually smaller compared to the multimodel ensemble spread. Our analysis and Leduc et al. (2019) found a substantial bias between the ClimEx LE and observational data aggregated to the 12 km grid. Especially over the Alps, a strong wet bias was identified. With regard to temperature, we found a cold bias for most grid points covering the Swiss Alps. The bias between model data and observations justifies the application of a bias adjustment procedure, which is explained in Sect. 2.4.

## 2.4 Bias adjustment

As simulations from general circulation models are usually too coarse to be directly used as input in impact models, CanESM2 was dynamically downscaled to a 12 km resolution, which better represents regional topography and therefore regional climatology. As practically all regional climate models still have a systematic model bias and as our impact model simulates snow for a particular one-dimensional point in space, another downscaling or bias adjustment step is needed to adjust systematic model biases and to bridge the gap between RCM simulations and the impact model. There exist multiple methods for bias adjustment and downscaling, and approaches are dependent on the study purpose. While, for example, the delta change approach is robust and easy to implement, the method neglects potential changes in variability and is therefore not suitable for our study. In recent years, many studies have concluded that quantile mapping (QM) performs similar or superior to other statistical

downscaling approaches (Feigenwinter et al., 2018; Gutiér-rez et al., 2019; Ivanov and Kotlarski, 2017).

When multimodel ensembles are bias-corrected, usually each simulation is corrected separately, whereas bias adjusting a single-model large ensemble requires specific considerations as the chosen method should not only correct the bias of each individual member but should also retain the individual intermember variability (Chen et al., 2019). We applied a distribution-based quantile mapping approach to bias-correct and downscale the model data to the station scale in one step, based on the daily translation method by Mpelasoka and Chiew (2009). Several studies tested this method and confirmed a reasonable performance (Gu et al., 2019; Teutschbein and Seibert, 2012). A downscaling-to-station approach was also performed for the official CH2018 Swiss climate scenarios (Feigenwinter et al., 2018). We modified the approach in the form that it was applied to a transient climate simulation ranging from 1980 to 2099, and, in contrast to previous studies that use daily scaling factors, it is based on subdaily scaling factors. The last-mentioned modification was necessary as SNOWPACK needs subdaily meteorological input.

In a first step, we computed the empirical distributions of observed and simulated climate variables for the baseline period 1984 to 2009. We chose this period as it is the period during which all of the stations have the smallest count of missing data. Overall, 99 quantiles were estimated separately for each month and each 3-hourly time step of the day. The grid cells overlying the respective climate stations described in Sect. 2.1 provide the simulated input series. For the ClimEx LE simulations, all 50 members were aggregated to compute a single empirical distribution. A large part of the internal variability would be removed or filtered if the distributions were calculated independently for each member (Chen et al., 2019). Thus, we computed the distribution based on the pooled ensemble members because all runs are derived from the same climate model with the same forcing and are therefore assumed to have the same climatological bias (Chen et al., 2019; Gu et al., 2019).

In a second step, we computed scaling factors between the simulated and observed data in the baseline period. For temperature, an additive scaling factor was estimated, while for the remaining variables multiplicative scaling factors were estimated. In a last step, for temperature the scaling factors were added to the empirical distributions of each single ensemble member, while for the remaining variables the scaling factors were multiplied with the empirical distributions of the simulations. The same scaling factors were transiently applied to the distribution functions of 25-year slices of each ensemble member.

The method does not take into account intervariable dependency. As SNOWPACK is a physically based model, physical inconsistencies in the data can lead to serious model errors. For example, precipitation occurring simultaneously with low humidity will result in model error. As we use a

univariate bias adjustment approach, we also had to test the results for these inconsistencies.

## 2.5 Validation of bias adjustment and SNOWPACK

The performance of the applied bias adjustment approach as well as the performance of SNOWPACK and the snow simulations using the ClimEx LE as a driver were validated prior to continued analyses.

To evaluate the performance of the bias adjustment approach, we compared the statistical characteristics of observed meteorological variables to the bias-adjusted ClimEx LE for subdaily, daily, monthly and yearly values and compared the diurnal and annual regimes of the corrected parameters. We obtained good results in the calibration period 1984 to 2009.

In terms of distributional quantities and climatologies, this is to be expected as the bias correction scheme was calibrated for exactly this period. Due to this reason and as the performance of the chosen bias correction method is already the subject of the studies by Chen et al. (2019) and Gu et al. (2019), we do not present the validation exercise in detail. Instead we focus on a validation that is highly relevant for snow accumulation, namely on the performance of the bias adjustment with regard to the snowfall fraction assuming the calibrated SNOWPACK rain–snow temperature thresholds. This is of special interest as a univariate bias adjustment approach, as employed here, does not explicitly correct for biased intervariable dependencies, which could, among other things, affect the snowfall fraction.

SNOWPACK itself was calibrated for the period 1985 to 1989 and cross-validated for different 5-year periods between 1990 and 2009 using observed meteorological input, which was compared to measured snow depths. All periods provide similar results. For a clearer visualization, we only show results for the period 2000 to 2004. To assess goodness of fit, we compared daily and monthly measured snow heights with modeled snow heights simulated with observed meteorological input using several performance indicators such as mean absolute error (MAE), Nash–Sutcliffe coefficiency CE5 (NSE), coefficient of determination ($R^2$ TS4) and index of agreement ($d$; e.g., Krause et al., 2005; Legates and McCabe, 1999).

## 2.6 Statistical analysis of simulated snow depth

In the first part of our analysis, we estimate the ensemble mean changes for annual mean and maximum snow depth between the reference period ranging from 1980 to 2009 (REF) and three future periods ranging from 2010 to 2039 (near future: FUT1), 2040 to 2069 (midfuture: FUT2) and 2070 to 2099 (far future: FUT3).

In the second part of the analysis, we estimate the uncertainties in future snow trends (and their drivers) emerging from ICV.

We apply the Mann–Kendall (MK) trend detection test (Kendall, 1975; Mann, 1945) to test for statistically significant trends of different lengths of time series starting in the year 2000 and ending between 2029 and 2099. The Mann–Kendall test is frequently used in climatological and hydrological applications (Gocic and Trajkovic, 2013; Kaushik et al., 2020). The MK test statistic ($S$) is computed as

$$S = \sum_{i=1}^{n-1} \sum_{j=i+1}^{n} \text{sgn}(x_j - x_i), \tag{1}$$

where $x_i$ and $x_j$ are the data points at times $i$ and $j$; $n$ represents the length of the time series; $\text{sgn}(x_j - x_i)$ is a sign function defined as

$$\text{sgn}(x_j - x_i) = \begin{cases} +1, \text{ if } x_j - x_i > 0 \\ 0, \text{ if } x_j - x_i = 0 \\ -1, \text{ if } x_j - x_i < 0. \end{cases} \tag{2}$$

The null hypothesis of the MK test is that a time series has no trend (Libiseller and Grimvall, 2002). In this study, trend significance was tested for $p$ values of 0.01, 0.05 and 0.1. As air temperature below a certain threshold occurring simultaneously with precipitation is the most important prerequisite of snowfall (Morán-Tejeda et al., 2013; Sospedra-Alfonso et al., 2015), we do not only test for future snow depth trends but also for the drivers of future snow conditions. For that reason, we also apply the Mann–Kendall test to temperature, precipitation, snowfall fraction and snowfall time series.

In the third part of the study, we estimate the change in IAV. As a measure for IAV, we use the standard deviations of annual mean and maximum snow depths for the respective reference and future periods. The standard deviation is commonly used as a measure of IAV, such as in Siam and Eltahir (2017). All analyses were performed with the statistical software R (R Core Team, 2017 CE6). All analyses in Sect. 3.3 to 3.5 are performed for mean and maximum winter snow depths. Winter is defined as the months October to March.

## 3 Results

## 3.1 Validation of SNOWPACK

The validation results of SNOWPACK using meteorological observations as input are summarized in Fig. 2. From Table 2, we can deduce a good model fit. For all stations, the annual snow regime is very similar between observations and simulations, and the month of maximum snow depth is always identical. Maxima are also well represented. There is no systematic over- or underestimation visible across stations. Only for the station Montana is a systematic overestimation of simulated snow depths across all years apparent.

The performance indicators of daily measured and simulated snow depths show good results for all stations. With

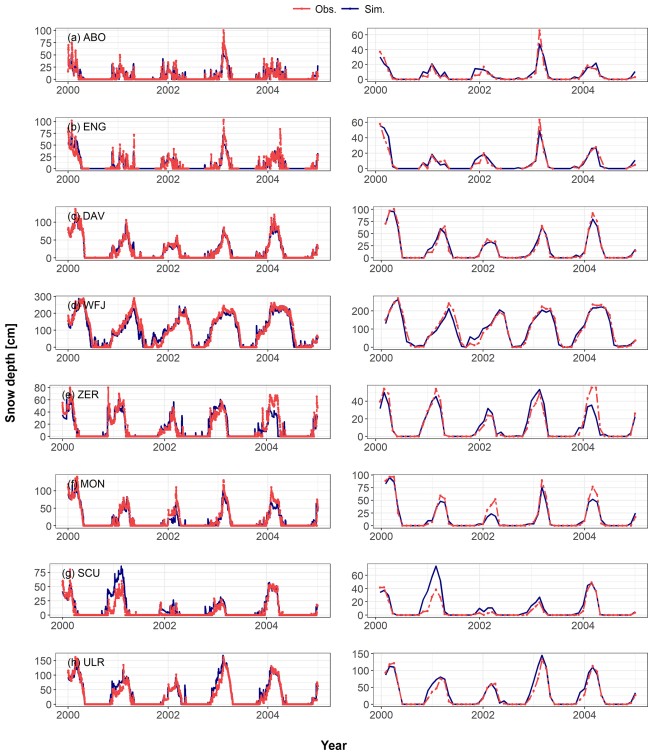

**Figure 2.** Validation results of SNOWPACK. Simulated daily (left) and monthly (right) mean snow depth driven by 3-hourly meteorological observations (sim) vs. measured daily and monthly mean snow depth (obs) for the period 2000 to 2004. See Table 1 for the abbreviations of case studies.

an $R^2$ and NSE larger than 0.75 and an index of agreement larger than 0.9, the stations Davos, Montana, Zermatt, Ulrichen and Weissfluhjoch show a very good model performance. The low-elevation stations Engelberg and Adelboden as well as Scuol still show a reasonably good model fit with an $R^2$ larger or equal to 0.6 and an NSE larger than 0.5 as well as an index of agreement ($d$) larger than 0.85.

### 3.2 Performance of bias adjustment and ensemble SNOWPACK simulations

Intervariable dependence between precipitation and humidity as well as incoming shortwave radiation was not corrupted through the bias correction (e.g., there were no cases of precipitation and simultaneously low humidity or precipitation and simultaneously high incoming shortwave radiation; not shown CE7). We concentrate the presentation of our results on the impacts of bias adjustment on temperature and precipitation dependencies and consequently snowfall fraction as this is an essential part of the snow modeling process.

Figure 3 visualizes the mean winter snowfall fraction for each station based on observed temperature and precipitation and based on the bias-adjusted temperature and precipitation data for the corresponding rainfall–snowfall thresh-

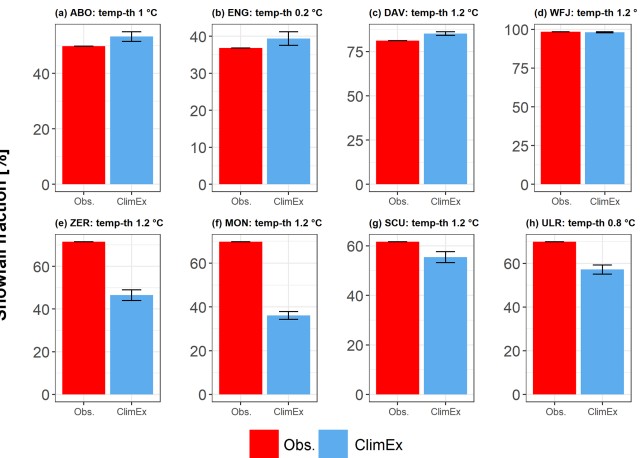

**Figure 3.** Mean winter snowfall fraction based on observations (obs; red) and for the bias-adjusted ClimEx LE (blue) for the period 1984 to 2009. *temp-th* indicates the calibrated snowfall–rainfall separation threshold for the respective station. See Table 1 for the abbreviations of case studies.

old temperatures. Snowfall fractions based on observations and bias-adjusted simulations are almost similar for Adelboden, Engelberg, Davos and Weissfluhjoch. For Scuol, the bias-adjusted ClimEx LE underestimates the observed snowfall fraction by 12 % and for Ulrichen by up to 16 %. For the two stations Zermatt and Montana, we find an even stronger underestimation of 33 % and 50 %, respectively. The results imply that there is no systematic error between observed and simulated snowfall fractions, but there are stations that show a significant underestimation of snowfall fraction compared to observations. The potential reasons for this underestimation are addressed in the discussion (Sect. 4).

In Fig. 4 we show the long-term monthly mean snow depths for observations, simulations driven by observed meteorological input and simulations driven by the bias-adjusted ClimEx LE. As already mentioned in Sect. 3.1, the comparison between observations and SNOWPACK simulations driven by observational data shows a good model fit for all stations. For the stations Adelboden, Engelberg, Davos, Weissfluhjoch and Scuol, the 50 ClimEx LE simulations enclose the observed snow depths for the calibration period of the bias adjustment. This implies a good performance of the bias adjustment. As shown above, the bias-adjusted data set systematically underestimates snowfall fraction in Zermatt and Montana, resulting in a pronounced underestimation of simulated snow depths. The two stations are not excluded from further analyses, but we have to clarify that those results must be interpreted cautiously and that we cannot consider absolute snowfall fraction or snow depth values for those stations.

Lastly, it is important to validate the ability of the ClimEx LE SNOWPACK simulations to reproduce IAV. In Fig. 5, we present IAV of winter mean snow depth for the ClimEx LE

**The Cryosphere, 14, 1–16, 2020** https://doi.org/10.5194/tc-14-1-2020

**Table 2.** Goodness-of-fit measures between measured daily snow depth and SNOWPACK simulated daily snow depth driven by 3-hourly meteorological observations for the period 2000 to 2004. Measures are mean absolute error (MAE), Nash–Sutcliffe coefficiency (NSE), coefficient of determination ($R^2$) and index of agreement ($d$). See Table 1 for the abbreviations of case studies.

|        | ABO  | ENG  | DAV  | WFJ  | ZER  | MON  | SCU  | ULR  |
|--------|------|------|------|------|------|------|------|------|
| MAE    | 3.91 | 5.75 | 4.47 | 18.5 | 4.58 | 5.66 | 4.49 | 7.21 |
| NSE    | 0.59 | 0.64 | 0.92 | 0.91 | 0.79 | 0.86 | 0.51 | 0.88 |
| $d$    | 0.87 | 0.9  | 0.98 | 0.98 | 0.94 | 0.96 | 0.9  | 0.97 |
| $R^2$  | 0.6  | 0.66 | 0.92 | 0.91 | 0.79 | 0.87 | 0.75 | 0.9  |

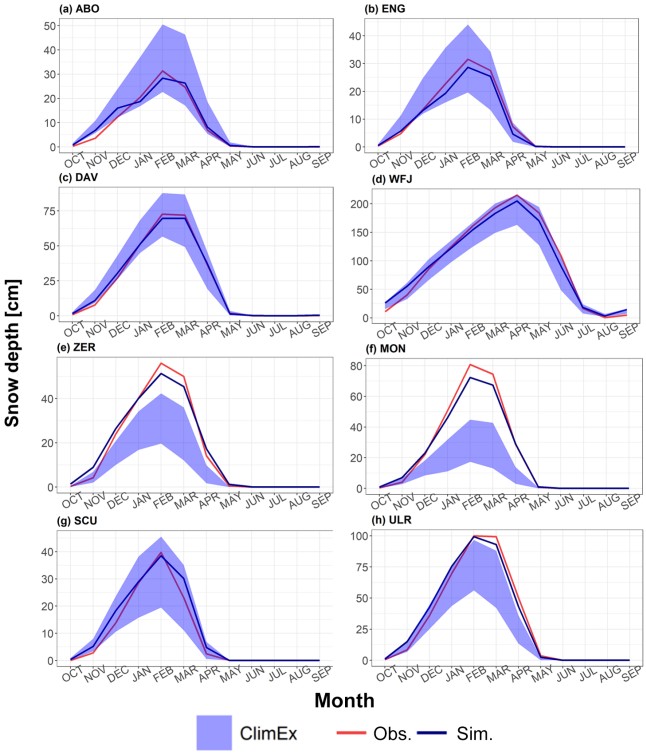

**Figure 4.** Long-term monthly mean snow depth for observations (obs), simulations driven by meteorological observations (sim) and the ensemble spread of simulations driven by the ClimEx LE for the period 1984 to 2009. See Table 1 for the abbreviations of case studies.

SNOWPACK simulations, observed snow depth and simulated snow depths based on observed meteorological input. For all stations but Zermatt and Montana, IAV of simulations driven by observational data lies within the envelope of the 50 ClimEx LE members. For the stations Zermatt and Montana, the systematic underestimation of snow depth consequently leads to an underestimation of IAV. Surprisingly, despite the satisfying validation results and in relation to the ClimEx LE spread, IAV between observations and simulations driven by observations differs significantly for the stations Adelboden, Engelberg and Scuol.

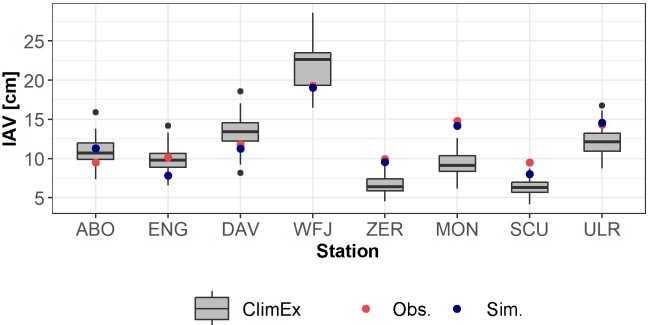

**Figure 5.** Interannual variability of mean winter snow depth for SNOWPACK simulations using ClimEx LE as a driver, using observed meteorological data as a driver (sim) and observed IAV (obs) of mean winter snow depth for the period 1984 to 2009. The boxes represent the interquartile range with the median as a horizontal line. The whiskers represent 1.5 times the interquartile range, and the dots represent outliers. See Table 1 for the abbreviations of case studies.

### 3.3 Mean climate change signal

Both the impacts of ICV on future snow trends and the changes in IAV under a given emission scenario must be put into perspective to the ensemble mean climate change signal. To do so, we present the absolute and relative changes in mean and maximum winter snow depth between our reference and future periods. Figure 6 visualizes the mean and maximum winter snow depths for the reference and future periods and the respective percentage change. For the ensemble mean, we find a continuous and significant decrease in mean and maximum snow depth over all stations. While absolute changes are partly stronger for maximum winter snow depths, the percentage decrease in mean winter snow depth is more severe for all case studies. All stations below 2000 m a.s.l. except Ulrichen show a similarly strong decrease in mean winter snow depth. The decrease in the near-future period is relatively small, and for all stations but Scuol, there is at least one member that simulates a small increase in winter mean and maximum snow depth. For the mid- and far-future periods, decreases range from −40 % to −60 % until 2069 and up to −60 % to −80 % until 2099. For Ulrichen, we obtain a slightly smaller decrease of up to −60 % until

2099. For the high-elevation station Weissfluhjoch, decreases are considerably lower, ranging from −30 % until 2069 to −50 % until 2099. The percentage decreases for maximum snow depths are considerably lower compared to those for mean snow depths, with an ensemble mean ranging from −30 % to −40 % until 2069 and −40 % to −60 % until 2099 for all stations but Weissfluhjoch. For Weissfluhjoch we observe ensemble mean decreases of −10 % until 2069 and −15 % until 2099. In the discussion, we compare these values to results from other studies CE8.

## 3.4 Significance of future snow depth trends

Despite the strong mean climate change signal described in Sect. 3.3, Sect. 3.4 points out the important role of ICV for the detection of statistically significant trends in future time series of snow depth and its most important drivers temperature, precipitation, snowfall fraction and snowfall. Note that, according to the poor validation results, we cannot draw conclusions on the absolute values of snowfall fraction, snowfall and snow depth for the stations Zermatt and Montana. Nevertheless, we can apply the trend test and compare relative changes. Figure 7 visualizes the results of the Mann–Kendall test for a positive trend in winter mean temperature and winter precipitation sums and a negative trend for winter snowfall fraction and snowfall as well as winter mean and maximum snow depth for the lowest station, Engelberg, and the highest station, Weissfluhjoch, for 1 %, 5 % and 10 % significance levels. Each time series starts in the year 2000 and ends in 5-year intervals between 2029 and 2099. Figure 8 shows the percentage of members with a significant Mann–Kendall test result for all case studies based on the 5 % significance level.

With regard to temperature, all case studies show a rapidly increasing percentage of significant positive trends. For a 30-year period, already between 45 % and 60 % of members show significant results. Until the year 2049 more than 90 % of members show a significant positive trend, and by 2059 all stations show a 100 % trend significance. Here we can clearly conclude that the anthropogenic climate change signal is significantly stronger than the ICV of temperature, i.e., the forced trend emerges from internal climate variability. For precipitation, we find a completely different picture. For all stations but Scuol, there is no clear sign towards an increase in future winter precipitation sums. There is a tendency towards an increasing number of members with a significant positive trend in precipitation towards the end of the century, but the percentage is below 50 % for all stations but Scuol. For Scuol, there is a clear sign towards an increase in winter precipitation sums. Here, more than 75 % of members show a significant positive trend in winter precipitation sums. In summary, we cannot detect a clear climate change signal for precipitation because of strong ICV.

With regard to snowfall fraction, we find a consistent increase in members with a significant negative trend in snow-

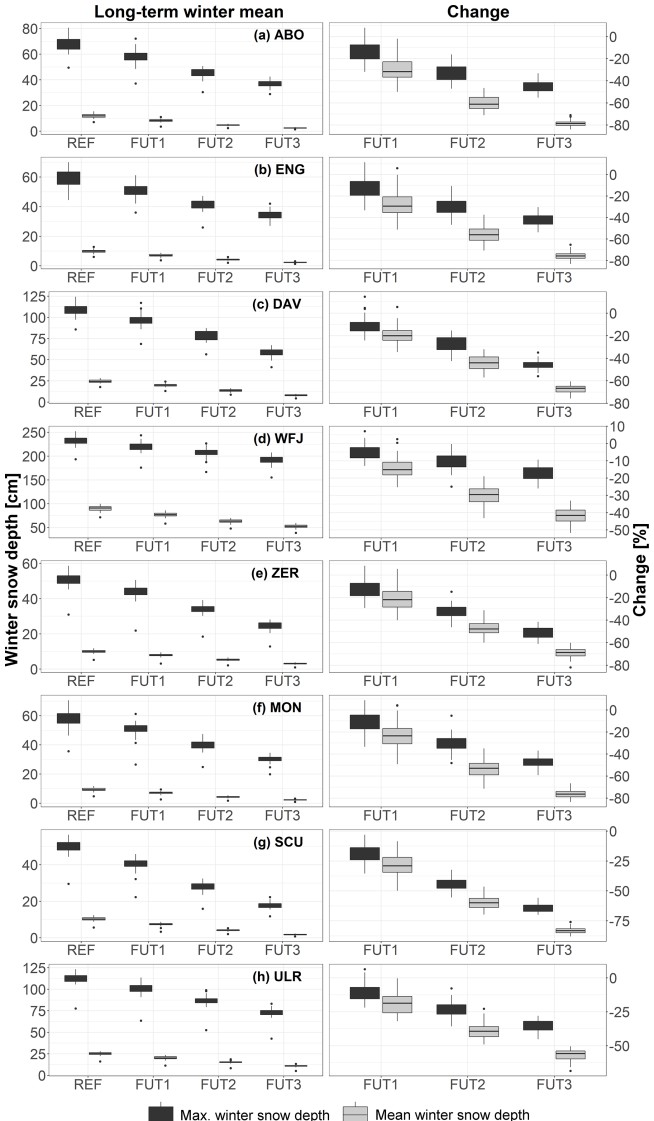

**Figure 6.** Winter mean maximum and mean snow depth CE9 (cm; left) for the ClimEx LE and mean changes (%; right) between the REF period 1980 to 2009 and FUT1 (2010 to 2039), FUT2 (2040 to 2069) and FUT3 (2070 to 2099) for the stations Adelboden (ABO), Engelberg (ENG), Davos (DAV), Weissfluhjoch (WFJ), Zermatt (ZER), Montana (MON), Scuol (SCU) and Ulrichen (ULR). The boxes represent the interquartile range with the median as a horizontal line. The whiskers represent 1.5 times the interquartile range, and the dots represent outliers.

fall fraction over time, but, as expected, the strong temperature signal does not translate into a similarly strong trend significance, and there are significant differences between case studies. Over a 50-year period, most stations show significance for only up to 50 % of members. The stations Davos, Weissfluhjoch and Scuol, the stations with the highest snowfall fraction in the reference period, show significant reductions for 75 % to 85 % of members over this period. This em-

phasizes the huge contribution of ICV to future trends in winter snowfall. Over a period of 60 years, there is still a 35 % chance of not detecting a significant negative trend in snowfall fraction for Montana due to ICV, but by 2069 all stations show significant decreases in snowfall fraction for more than 90 % of members.

In contrast to snowfall fraction, we find a lower statistical significance for negative trends of total winter snowfall sums. For all stations except Weissfluhjoch, over a period of 60 years, the percentage of statistically significant negative trends in winter snowfall sums is only between 25 % and 60 %. Over a period of 80 years between 60 % and 90 % of members show a statistically negative trend, and by the end of the century only the stations Engelberg, Zermatt, Montana and Scuol show a trend significance for 100 % of members. For the station Weissfluhjoch, no statistically significant negative trend is obtained over any period. Even over 100 years less than 10 % of members show a significant reduction in winter snowfall sums. These results imply that, despite strong temperature increase and a significant reduction in snowfall fraction, a reduction in total snowfall sums at this site remains very uncertain.

Similar to snowfall fraction, we find a steady increase in the count of members with a statistically significant negative trend in winter mean snow depth, but compared to winter temperature, this development starts considerably later. Generally, the trend significance is stronger compared to snowfall fraction, but there is still a 50 % to 25 % chance (for all stations but Scuol, where more than 90 % of members show significant negative trends) that ICV will superimpose anthropogenic climate change impacts on mean winter snow depth over a period of 50 years. By 2069, the percentage of significant members increases to more than 90 %. The stronger significance of mean snow depth compared to snowfall fraction and the higher significance compared to snowfall sums imply the combined effects of a very uncertain decrease in snowfall sums combined with more rapid and more frequent snow melt. Lastly, maximum winter snow depth shows a significantly different evolution than mean winter snow depth. Here, we also obtain large differences between the lower-lying stations and the highest station, Weissfluhjoch. For most CE10 stations but Scuol, there is a probability of more than 50 % of no significant negative trend in future maximum snow depth over a period of 50 years. For Scuol this probability is only 15 %, while it is 80 % for Weissfluhjoch. Over a period of 80 years all stations but Weissfluhjoch show a significant decrease in maximum snow depths in more than 90 % of the cases. For Weissfluhjoch a high probability for ICV to superimpose anthropogenic climate change remains. By 2049 the percentage of negative trends is below 20 %, by 2079 the probability is still below 50 %, and even over a period of 100 years there is a 25 % chance of no significant negative trend in maximum winter snow depth. This emphasizes that also in the far future, individual important snow peaks can be expected, especially at high elevations.

Our results underline the outstanding contribution of ICV to uncertainties related to future trends in snow depth and its drivers. Especially in the near-future, ICV can hamper a clear impact signal of anthropogenic warming as the strong signal for mean winter temperature does not directly translate into clear snow-related signals.

## 3.5 Changes in interannual variability

In Sect. 3.4 we revealed the large contribution of ICV to uncertainties related to future trends in snow depth. However, the variability of snow depth itself is likely to change with anthropogenic forcing. Here we investigate how IAV of mean and maximum snow depth, defined as the standard deviation of snow depth over a period of 30 years, is likely to change under RCP8.5. From Figs. 6 and 9 we can see that a gradual decrease in winter mean and maximum snow depth is accompanied by a decrease in absolute IAV. Nevertheless, in relative terms (relative to the mean of the corresponding periods), IAV can strongly increase in the future, but there are differences between mean and maximum snow depth and at different stations. In the reference period and at lower elevation, relative IAV of mean snow depth lies between 30 % (Scuol) and 70 % (Engelberg), and relative IAV of maximum snow depth lies between 20 % (Scuol) and 60 % (Engelberg). For most cases, relative IAV of mean snow depth is larger compared to maximum snow depth. For Weissfluhjoch the overall variability is lower compared to the other stations (22 % in the reference period), and maximum snow depth has a larger variability than mean snow depth. For Weissfluhjoch, an increase in relative IAV can be found for neither mean nor maximum snow depth. In contrast, an increase in relative IAV for the stations Adelboden, Engelberg, Davos, Zermatt, Montana, Scuol and Ulrichen is projected. Larger increases in relative IAV are obtained for mean winter snow depth, while the increases for maximum snow depth are very small. For Davos, for example, we find an ensemble mean increase in relative IAV of mean snow depth from 35 % to 55 % until the end of the century. Increases in relative IAV of maximum snow depth range from 35 % to 40 %. For Montana IAV increases from 50 % in the reference period to more than 80 % in the future 2 period. For Scuol we find an increase from 40 % up to 70 % between the two periods.

## 4 Discussion

Many studies have significantly improved our knowledge about the cryosphere in a future climate (Barnett et al., 2005; Beniston et al., 2018). Nevertheless, the predominant number of studies focuses on changes in the mean, while studies on the interdependencies of climate change and ICV are very rare. Our analysis is the first study that uses a single-model large ensemble as input for a physically based snow model. This allows a probabilistic uncertainty assessment of future

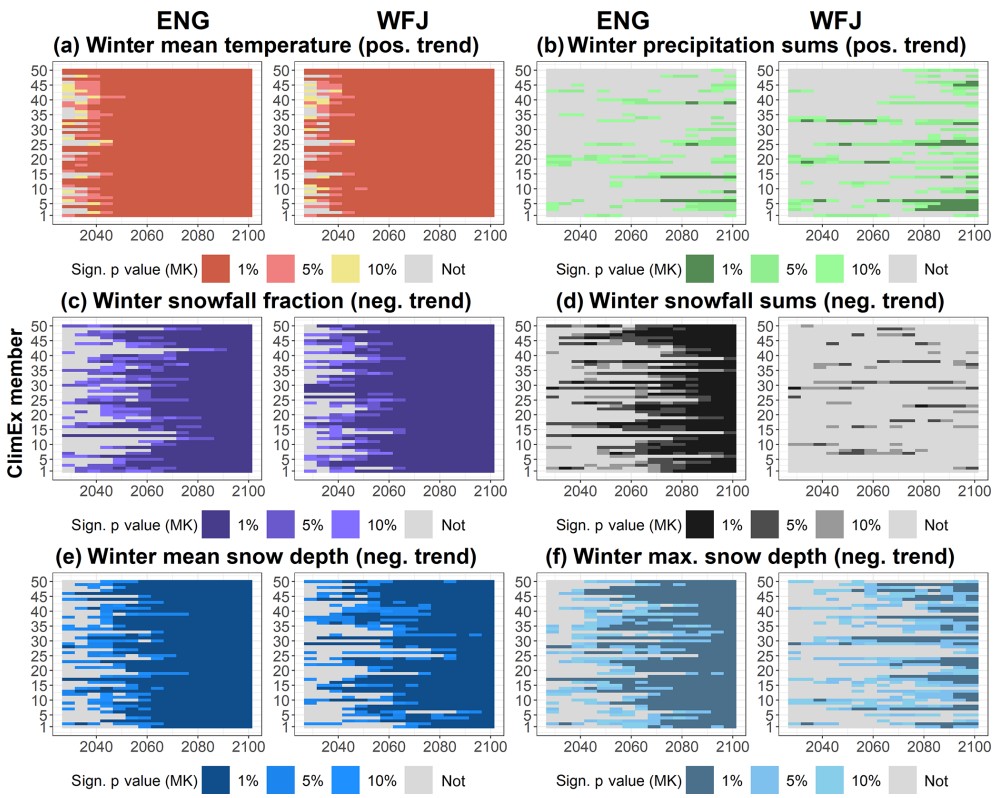

**Figure 7.** Heat maps of trend significance for the Mann–Kendall test for different periods starting in 2000 and ending between 2029 and 2099 for winter mean temperature, winter precipitation sums, winter snowfall fraction, winter snowfall sums, and winter mean and maximum snow depth for the lowest station, Engelberg (ENG), and the highest station, Weissfluhjoch (WFJ).

snow trends in the European Alps attributable to ICV. We further estimate how IAV might change in a future climate.

### 4.1 Uncertainties and limitations

While we are gaining important insights into the dynamics of mean, maximum and interannual variability of snow depth and the role of ICV under climate change conditions, a number of important uncertainties and limitations must be taken into account, which span over the whole modeling process. Important boundary conditions are that our results are highly dependent on the choice of the emission scenario and the combination of global climate model (GCM) and RCM CE11 as well as the selected bias adjustment approach. First, it must be stated that the ClimEx LE is still unique regarding members and spatiotemporal resolution; like other currently existing single-RCM initial-condition large ensembles, it is available under RCP8.5 only. Being aware of the extreme character of this GHG CE12 concentration scenario and the high sensitivity of the GCM–RCM combination, the results obtained from the presented analyses are considered valid as they represent the expected dynamics and states of other emission scenarios but reach certain levels of change earlier in time.

Due to the single-model approach, it is understood that the presented setup has limited capacity in providing a robust estimate of anthropogenic climate change; this is where multimodel ensemble setups have clear advantages (Tebaldi and Knutti, 2007). Comparing the detected climate change signals of the ClimEx LE with the EURO-CORDEX ensemble shows that the data used for this study provide a highly sensitive forced response, yet within plausible ranges (von Trentini et al., 2019). Nevertheless, multimodel ensembles make it very difficult to distinguish between model uncertainties and ICV, which is a major advantage of our approach, when the goal is to study ICV. Of course, it would be of interest to estimate model uncertainties and do a probabilistic analysis of ICV. To do so an ensemble of ensembles would be the preferred approach. Due to computational limitations, such analyses are not yet feasible, especially on the regional scale.

Another source of uncertainty is the choice of the bias adjustment methodology. While quantile mapping was found to be similar or superior to many other bias adjustment approaches (Gutiérrez et al., 2019; Ivanov and Kotlarski, 2017; Teutschbein and Seibert, 2012), it has some important drawbacks. QM assumes stationarity of the model bias structure, an assumption that is uncertain under changing climatic conditions (Maraun, 2013). Furthermore, QM cannot cor-

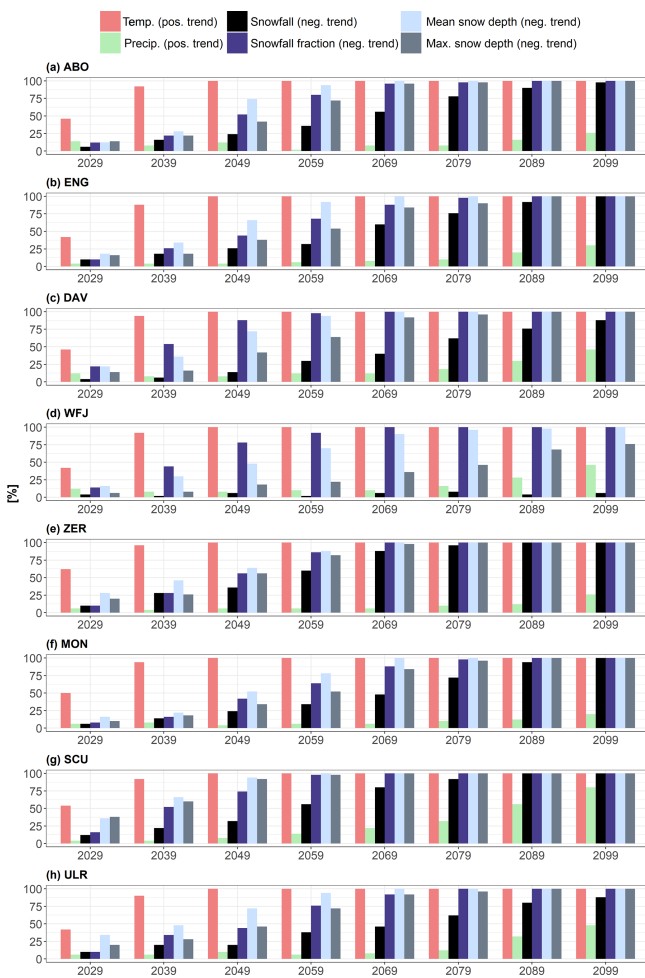

**Figure 8.** Percentage of significant trends (*p* value of 5 %) for time series of temperature, winter precipitation sums, snowfall fraction, snowfall sums, and mean and maximum snow depth starting in 2000 and ending between 2029 and 2099 for all case studies. See Table 1 for the abbreviations of case studies.

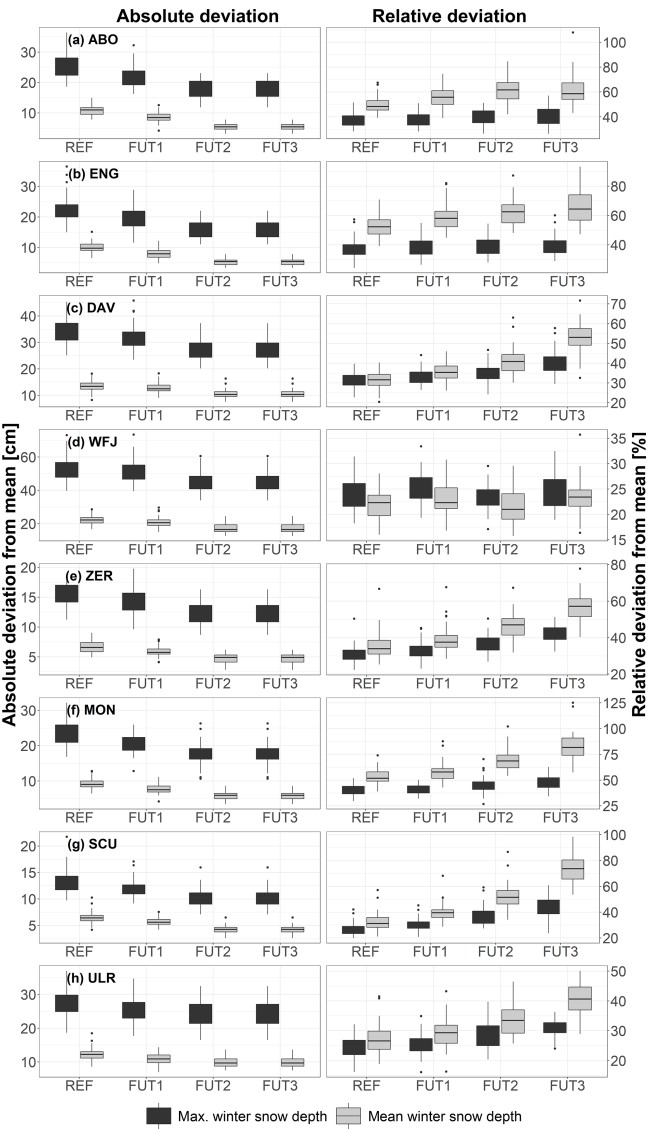

**Figure 9.** Interannual variability for the REF (1980 to 2009), FUT1 (2010 to 2039), FUT2 (2040 to 2069) and FUT3 (2070 to 2099) periods expressed as the absolute standard deviation (cm; left) and standard deviation relative to the mean (%; right) for mean and maximum snow depth for the stations Adelboden (ABO), Engelberg (ENG), Davos (DAV), Weissfluhjoch (WFJ), Zermatt (ZER), Montana (MON), Scuol (SCU) and Ulrichen (ULR). The boxes represent the interquartile range with the median as a horizontal line. The whiskers represent 1.5 times the interquartile range, and the dots represent outliers.

rect misrepresented temporal variability (Addor and Seibert, 2014). Therefore, interannual variability was validated in Sect. 3.2, yielding acceptable results. When applied in a downscaling context, QM cannot reproduce local processes and feedbacks as QM is a purely empirical approach (Feigenwinter et al., 2018; Kotlarski et al., 2015). In contrast, QM can modify the raw climate change signal and simulated trends (Ivanov et al., 2018). This point is especially important in our study as we have to correct each member based on the empirical distribution of the whole ensemble to retain the internal climate variability. Therefore, the climate change signals of the single members are modified. Cannon et al. (2015) developed a method that preserves the raw climate change signal, but applied to this study it would only preserve the ensemble mean signal. Further research is needed to develop potential methods that preserve the climate change signal for single members from single-model ensem-

bles. As a last point, we employ a univariate bias adjustment approach, which treats all meteorological variables independently. While intervariable consistency cannot be guaranteed (Feigenwinter et al., 2018), multiple studies show that QM generally maintains intervariable consistency (Ivanov and Kotlarski, 2017). In the course of this work, intervariable consistency was validated, and we obtained good results for

radiation, precipitation and humidity. Variable consistency with regard to snowfall fraction was inaccurate for individual case studies. Prior to bias adjustment, a strong cold bias over most grid points caused snowfall fraction to be significantly too high. Therefore, bias correction generally improved the simulated snowfall fractions. The exploration of possible reasons for inaccurate snowfall fractions in some cases will be subject to future work.

Schlögl et al. (2016) found that the uncertainties from the snow model itself account for approximately 15 %. However, as we focus on investigating relative changes in snow depth, this source of uncertainty is of less concern.

Lastly, as most stations are situated in elevations between 1320 m a.s.l. and 1640 m a.s.l., a detailed analysis on elevation dependencies could not be performed.

## 4.2    Discussion of results in the context of existing research and potential future research

Despite the above-mentioned uncertainties and limitations, this study can provide important insights into the interdependencies between anthropogenic climate change and ICV and its impacts on snow depth in the Alps. Its novelty stems from a true probabilistic assessment of ICV. In the first part of our results section, we presented the ensemble mean change between a reference period (1980 to 2009) and three future periods and found significant decreases in ensemble mean snow depth in the future. Schmucki et al. (2015) present a similar analysis for partly the same case studies using 10 GCM–RCM model chains from the ENSEMBLES project under the IPCC A1B emission scenario. Although the reference periods do not fully match (Schmucki et al., 2015, use 1984 to 2010), we can put the changes between the reference period and the mid- (2040 to 2069 in this study and 2045 to 2074 for Schmucki et al., 2015) and far-future period (2070 to 2099) into perspective. For Weissfluhjoch, Schmucki et al. (2015) simulate a mean decrease of 28 % (near future) and 35 % (far future), which is close to our simulation results of CE13 −20 % (near future) and −29 % (far future) in mean winter snow depth and −30 % (near future) and −42 % (far future) for annual mean snow depth. Both studies show comparable decreases in mean snow depth at Weissfluhjoch, although our study uses the much stronger RCP8.5 compared to the A1B scenario in Schmucki et al. (2015).

In the near and far future, Schmucki et al. (2015) found an ensemble spread of mean snow depth of 35–135 cm (near future) and 30–130 cm (far future), whereas we simulate an ensemble spread of 62–87 cm (near future) and 52–82 cm (far future) for winter mean snow depth and 48–70 cm (near future) and 39–60 cm (far future) for annual mean snow depth.

Marty et al. (2017) use 20 different GCM–RCM chains to compare the impacts of different emission scenarios on mean snow depth for two catchments in Switzerland that partly also cover stations analyzed in this study. For the Aare catchment that covers our stations Adelboden and Engelberg and under

IPCC A2, Marty et al. (2017) simulate an ensemble mean decrease of 65 % and an ensemble spread between −33 % and −85 % between the reference period (1999 to 2012) and the far future (2070 to 2099). For our case studies Adelboden and Engelberg, we find decreases in mean snow depth between 65 % and 80 % in the far future. Put into a global perspective, Kudo et al. (2017) investigate the uncertainties in future snow projections related to GCM uncertainties in Japan and find snow equivalent reductions between 65 % and 90 % based on 11 climate projections derived from five GCMs.

Accordingly, the ensemble spread of the single-model large ensemble (present work) is considerably smaller compared to previous assessments based on multimodel ensembles. This agrees with results by von Trentini et al. (2019), who found that for temperature and precipitation, the single-model spread is usually smaller compared to the multimodel ensemble. The results emphasize the large uncertainties related to the choice of the GCM–RCM model chain that could be mistaken and falsely interpreted as ICV.

While a regression-based analysis of different elevations is not possible due to the limited elevation ranges, we still find significant differences between stations, especially between the highest station, Weissfluhjoch, and the lower-elevation case studies. With regard to trend significance, we can conclude that for all stations there is a nonnegligible probability of hiatus periods of mean and maximum snow depth of lengths up to 50 years. Still, those probabilities are highest for Weissfluhjoch, where we find a probability of more than 50 % that there will be no significant reduction in future maximum winter snow depth over a period of 80 years. This is also confirmed by Morán-Tejeda et al. (2013) and Kudo et al. (2017), who find different drivers for changes in snowpack and different responses to anthropogenic warming for different elevation bands.

We also find an uneven response of different snow metrics to anthropogenic warming. Statistically significant trends are first detected for mean winter snow depth, followed by winter snowfall fraction and later still by winter maximum snow depth; for trends in winter snowfall sums, we can identify large uncertainties related to ICV. Our results are confirmed by Ishida et al. (2019) for three case studies in California; they investigate climate change impacts on interrelations between snow-related variables and find that temperature rise will affect but will not dominate the future change in snow water equivalent and also find uneven responses of different snow-related variables to anthropogenic forcing. Further, these results coincide with Pierce and Cayan (2013) and emphasize two points. First, also in the far future, we must expect considerable winter snowfall sums and events of large snow accumulation, even under RCP8.5. Overall, a reduction in mean snow depth is rather driven by increased snowmelt than by a strong decrease in absolute snowfall sums. Consequently, trend detections for maximum snow depths over periods of fewer than 50 years largely depend on noise from ICV. Second, ICV remains the highest source of uncertainty

over a short to medium range of time, but it can even hamper a statistically significant signal over periods of more than 50 years. On the other hand, with regard to future research, ICV cannot only reveal the possibilities of long hiatus periods, but it can also illustrate even faster snowpack declines in the Swiss Alps.

In this study, we found that ICV does not only obscure the forced climate change signals but that variability in terms of IAV itself is likely to change in the future. These findings do not only support our understanding of the ranges of internal climate variability, they are particularly useful to distinguish the "noise" of climate variability from "real" climate change signals. With regard to changes in IAV, Weissfluhjoch is the only station where we cannot identify a change in the IAV relative to the mean. For the remaining stations, we find that anthropogenic climate change has an impact on IAV. A thorough investigation of the causes of this change is beyond the scope of the present work. We assume that snow-rich and snow-scarce winters are often dependent on general circulations, such as large-scale blockings (García-Herrera and Barriopedro, 2006), and also large-scale oscillations caused by El Niño or the Arctic Oscillation (Seager et al., 2010; Xu et al., 2019). These factors remain insufficiently studied, and their identification could be the subject of future research that takes into account large-scale synoptic patterns from the ClimEx LE.

## 5 Summary and conclusions

In the present work, we analyzed the interdependencies between ICV and anthropogenic climate change and its impacts on snow depth for eight case studies across the Swiss Alps. For this purpose, we made use of a 50-member single-model RCM ensemble and used it as a driver for the physically based snow model SNOWPACK. The large number of members used in this study allowed for a probabilistic analysis of ICV. We can confirm our first hypothesis, which states that ICV is a major source of uncertainty in trends of future Alpine snow depth. By applying a Mann–Kendall trend test, we estimate the trend significance of snow depth and its main drivers for time series of different lengths (i.e., different lead times). We present the probabilities of detecting significant trends caused by anthropogenic forcing in the presence of ICV and find that ICV is a major source of uncertainty for lead times up to 50 years and more.

We can also confirm our second hypothesis, which states that IAV of snow depth will change with anthropogenic climate forcing. To answer our initial research question, we compare interannual variabilities of snow between a reference period and three future periods and find that, relative to the mean, IAV of snow considerably increases in the future for most CE14 cases but the high-elevation station Weissfluhjoch.

Our results show how important it is to not only analyze changes in the mean snow depth but also its variability as it is a dominant source of uncertainty and because variability itself can change with anthropogenic climate change. For all economies that are directly dependent on snow or runoff from snowmelt, future climate impact assessments are hence subject to important uncertainties. On the one hand, climate change will significantly reduce snow cover, but the extent remains disputed, and ICV is one of the top sources of this uncertainty. On the other hand, in addition to a reduction in mean snow depths, its variability is likely to change. This will additionally increase vulnerabilities of snow-dependent economies in the future.

*Data availability.* The ClimEx LE data analyzed in this study are publicly available via the ClimEx project web page (https://www.climex-project.org/en/data-access TS5). The observational data sets as well as the snow depth data for Switzerland are available for research and educational purposes via the IDAweb by MeteoSwiss (https://www.meteoswiss. admin.ch/home/services-and-publications/beratung-und-service/ datenportal-fuer-lehre-und-forschung.html TS6). On request, the analysis code is available from the corresponding author.

*Author contributions.* All authors designed the research. FW performed the simulations and analyzed the data. FW wrote the draft. All authors edited the paper.

*Financial support.* This research has been supported by the NAME OF FUNDER (grant no. GRANT AGREEMENT NO). TS7

*Competing interests.* The authors declare that they have no conflict of interest.

*Acknowledgements.* We would like to thank Pirmin Ebner and Mathias Bavay from the Institute of Snow and Avalanche Research (SLF) for technical support in SNOWPACK. We would like to thank Christoph Marty (SLF) for support in the research design. Further, we would like to thank Magdalena Mittermeier, Fabian v. Trentini and Raul Wood from Ludwig-Maximilians-University Munich for their help in the processing of the ClimEx LE raw data. This work was supported by an ETH Zurich ISTP Research Incubator grant.

*Review statement.* This paper was edited by Jürg Schweizer and reviewed by two anonymous referees.

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

## Remarks from the language copy-editor

CE1    Should this be defined?
CE2    Should this be defined?
CE3    Do you mean Shakoor et al. and Shakoor and Ejaz used the Konzelmann parameterization?
CE4    Should this be Canadian Regional Climate Model?
CE5    Should this be efficiency coefficient?
CE6    This reference is not in the list.
CE7    What does this refer to?
CE8    Please confirm the change.
CE9    Do you mean "winter mean and maximum snow depth"?
CE10   Do you mean "For all stations but Scuol"?
CE11   Please confirm the change.
CE12   Please define.
CE13   Should these have the same sign as the Schmucki et al. findings?
CE14   Do you mean "for all cases but the high-elevation station"?

## Remarks from the typesetter

TS1    Please provide department if possible.
TS2    The composition of Figs. 2–9 has been adjusted to our language standards.
TS3    Please provide date of last access (dd/mm/yyyy).
TS4    Please confirm exponential format here and throughout the text.
TS5    Please provide a direct link to the data set, if possible. In any case, please provide a reference list entry including creators, title, and date of last access.
TS6    Please provide a direct link to the data set, if possible. In any case, please provide a reference list entry including creators, title, and date of last access.
TS7    Please note that there is funding information given in the acknowledgements, but you did not indicate any funding upon manuscript registration. Therefore, we were not able to complete the financial support statement. Please provide the missing information and double-check your acknowledgements to see whether repeated information can be removed from the acknowledgements. Thanks.
TS8    Please provide pages or article number.
TS9    Please check asteriks.
TS10   Please provide pages.
TS11   Please provide pages.
TS12   Please provide pages.
TS13   Please provide more information like pages or URL link with last access date.
TS14   Please provide pages.
TS15   Please provide pages.
TS16   Please provide pages.
TS17   Please provide volume and pages.