# Peer review of "Anthropogenic climate change versus internal climate variability: Impacts on snow cover in the Swiss Alps"

_The Cryosphere, 2020_

## Referee Comment (RC1) · Anonymous Referee #1 · 30 Apr 2020

**General comments:**

The authors present a very relevant manuscript about the impacts of climate change and internal climate variability on snow cover at 8 stations located in the Swiss Alps. This is one of the few studies adressing the role played by internal climate variability in the future evolution of snow properties. The methods are sound and the manuscript is well-written. I thus have only minor comments outlined below.

**Specific comments:**

- Title: As the study focusses on 8 stations all located in the Swiss Alps, I think this should be clearly stated in the title of the manuscript. Currently, the title misleads the reader into thinking that the study encompasses the impacts on snow over the whole

Alps.

- Most figures and tables could benefit from a more explicit caption, to make sure that they are self-explanatory. For example, Figs. 2-5 and 8, and Table 2 should mention something like "See Table 1 for the acronyms of case studies". Additionally, the caption of Table 2 should remind the reader what "MAE", "NSE", "d" and "R2" are. In Figs. 5-6 and 9, the meaning of the black dots (outliers?) is not given in the caption, nor any explanation about how to read the box-and-whisker plots (those are standard representations, but still a short sentence to explain them could greatly improve the comprehension of the figures). THe caption of Fig. 6 should be changed to: "Winter mean maximum and mean snow depth...".

- As a general recommendation, please avoid double parentheses (for example, line 69, or line 209, ...).

- Line 379-380: what do the percentages inside parentheses refer to? Scuol? If so, the sentence should be rephrased: "For Montana (Scuol), IAV increases from 50% (40%) ... up to more than 80% (70%)...".

- Section 4.2: I understand that it is convenient to compare results of this study to another study (Schmucki et al., 2015) that focussed on the same stations. However, would it be possible to have some general comparison with other alpine studies outside of Switzerland?

**Technical corrections:**

- Line 27: Please rephrase to: "In large parts of the world, water stored...".

- Line 46: "Typically, studies compare different emission scenarios".

- Line 73: "the following hypotheses".

- Line 78: please remove the comma between "a" and "dynamically".

- Line 111: "the lack of".

- Line 131: "The ClimEx LE consists of".

- Line 196: Please consider splitting this sentence into two after "parameters".

- Line 212-213: "from 1980 to 2009... from 2010 to 2039..., 2040 to 2069 ... and 2070 to 2099".

- Line 266-267 (caption of Fig. 3): please replace the comma by a full stop.

- Line 274: please remove the comma after "clarify".

- Line 291: please remove the comma after "Both".

- Fig. 6: the title of the left column should read "Long-term winter mean".

- Line 400: there seems to be a word missing before "just reach". Maybe "but"?

- Line 422: please remove the comma after "While".

- Line 436: "we presented".

- Line 446: please remove the comma after "found".

- Line 446-447: please replace the two occurrences of the word "between" by the word "of".

- Line 470: please remove the comma after "station".

- Line 472: "the present work".

- References: I noted a typo at line 676. Please check all references for other typos.
* * *

---

## Referee Comment (RC2) · Anonymous Referee #2 · 14 May 2020

General comments

This paper presents the analysis of climate change trends of different variables related to the snow cover. This analysis includes data coming from a rare (if not unique) high resolution RCM large ensemble. It explores the inclusion of the Internal Climate Variability (ICV), available with such an ensemble, into the analysis of the trends of the variables and their significance. The paper also takes quick a look at the effect of climate change on the Inter Annual Variability (IAV). The analysis is done for 6 Swiss alpine stations and uses bias corrected RCMs fields used to drive the SNOWPAK snow model to obtain the snow-related variables.

I really enjoyed this paper. I thought it was very well done and I recommend its publication as I think it contributes to the general knowledge about the effect of climate change

on snow.

Specific Comments

In the Introduction, the authors write about "downscaled model" but they do not specify that it is a Regional Climate Model (RCM). I think it should be corrected. In fact, RCM appears first on line 58 and is not defined.

On line 72 it is written that "Again, a single model large ensemble can help answer how inter-annual variability might respond to changes in climatic forcing". I do not disagree but a single model simulation could do the trick or an ensemble of one member simulations of different RCMs could also do the trick. This ensemble does not strike me as ideal for that task.

On line 184 the authors write, "The same scaling factors are transiently applied to the distribution functions of 25-year slices of each ensemble member and pushed into the future by a ten year moving window." I am not sure I understand what it means.

On figure 2, is it me or the colours of the lines are inverted from left to right (i.e. blue on left is red on right)? I also wonder what the monthly mean figures add to the daily ones.

For figure 7, I did not understand why the variable presented is snowfall fraction instead of snowfall. I understand the presentation of the validation of SNOWPAK is done with that variable but we have opposed trends for snowfall fraction and a signal for the total precipitation and we are interested into the mean and max snow depth. It seems to me it would be natural to look at total snowfall trends. I also think the sign of the trend should be visible on the figure or at least in the legend. For now we have to go look into the text to see if the trend is positive or negative.

For section 3.5, it was not clear to me if the mean used to compute the relative IAV is the mean of the reference period or the mean of every period. Generally speaking, this section was interesting but a bit light and unclear at the same time. I also think the

relation between ICV and IAV could be better explored.

On line 368 it is written "From Fig. 6 and Fig. 9 we can see that a gradual decrease of winter mean and maximum snow depth leads to a decrease in absolute IAV". What does this mean? I have problems with the "leads" word.

Line 414: "temporal variability was validated in Sect. 3.1". That is not clear to me, Sect. 3.1 presented the validation of snow depth time series for SNOWPAK (fig. 2), there is no question/measure of temporal variability, no?

Technical corrections

I do not have any purely technical corrections to report.
* * *

---

## Author Comment (AC1) · 2 Jun 2020

We thank all anonymous reviewers for their time and effort and their helpful and constructive comments. The original comments of the reviewers are in blue. Our replies are in black.

**Anonymous Referee #1:**
**General comments:**
    1.   The authors present a very relevant manuscript about the impacts of climate change and internal climate variability on snow cover at 8 stations located in the Swiss Alps. This is one of the few studies adressing the role played by internal climate variability in the future evolution of snow properties. The methods are sound and the manuscript is well-written. I thus have only minor comments outlined below.

>>>> We thank referee 1 for the very positive general comments and helpful specific comments. We assiduously went through the comments and complemented and refined the manuscript accordingly.

**Specific comments:**
    2.   - Title: As the study focusses on 8 stations all located in the Swiss Alps, I think this should be clearly stated in the title of the manuscript. Currently, the title misleads the reader into thinking that the study encompasses the impacts on snow over the whole Alps.

>>>> We changed the title to: *Anthropogenic climate change versus internal climate variability: Impacts on snow cover in the Swiss Alps*

    3.   - Most figures and tables could benefit from a more explicit caption, to make sure that they are self-explanatory. For example, Figs. 2-5 and 8, and Table 2 should mention something like "See Table 1 for the acronyms of case studies". Additionally, the caption of Table 2 should remind the reader what "MAE", "NSE", "d" and "R2" are. In Figs. 5-6 and 9, the meaning of the black dots (outliers?) is not given in the caption, nor any explanation about how to read the box-and-whisker plots (those are standard representations, but still a short sentence to explain them could greatly improve the comprehension of the figures). THe caption of Fig. 6 should be changed to: "Winter mean maximum and mean snow depth...".

>>>> We refined the figure and table captions to:
*Figure 2: Validation results of SNOWPACK. Simulated daily (left) and monthly (right) mean snow depth driven by 3-hourly meteorological observations (sim) vs. measured daily and monthly mean snow depth (obs) for the period 2000 to 2004.  See Table 1 for the acronyms of case studies.*

*Table 2: Goodness of fit measures between measured daily snow depth and SNOWPACK simulated daily snow depth driven by 3-hourly meteorological observations for the period 2000 to 2004. Measures are: mean absolute error (MAE), Nash-Sutcliffe coefficiency (NSC), coefficient of determination (R2) and index of agreement (d). See Table 1 for the acronyms of case studies.*

*Figure 3:  Mean winter snowfall fraction based on observations (obs; red) and for the bias-adjusted ClimEx LE (blue) for the period 1984 to 2009. temp-th indicates the calibrated snowfall-rainfall separation threshold for the respective station. See Table 1 for the acronyms of case studies.*

*Figure 4: Long-term monthly mean snow depth for observations (obs), simulations driven by meteorological observations (sim) and the ensemble spread of simulations driven by the ClimEx LE for the period 1984 to 2009. See Table 1 for the acronyms of case studies.*

*Figure 5: Inter-annual variability of mean winter snow depth for SNOWPACK simulations using ClimEx LE as driver, using observed meteorological data as driver (sim) and observed IAV (obs) of mean winter snow depth for the period 1984 to 2009. The boxes represent the inter-quartile range with the median as horizontal line. The whiskers represent 1.5 times the inter-quartile range and the dots represent outliers. See Table 1 for the acronyms of case studies.*

*Figure 6: Winter mean maximum and mean snow depth [cm] (left) for the ClimEx LE and mean changes [%] (right) between the REF period 1980 to 2009 and FUT1 (2010 to 2039), FUT2 (2040 to 2069) and FUT3 (2070 to 2099) for stations Adelboden (ABO), Engelberg (ENG), Davos (DAV), Weissfluhjoch (WFJ), Zermatt (ZER), Montana (MON), Scuol (SCU) and Ulrichen (ULR). The boxes represent the inter-quartile range with the median as horizontal line. The whiskers represent 1.5 times the inter-quartile range and the dots represent outliers.*

*Figure 7: Heat maps of trend significance for the Mann-Kendall test for different periods starting in 2000 and ending between 2029 and 2099, for winter mean temperature, winter precipitation sums, winter snowfall fraction, winter snowfall sums and winter mean and maximum snow depths, for the lowest station Engelberg (ENG) and the highest station Weissfluhjoch (WFJ).*

*Figure 8: Percentage of significant trends (p-value 5%) for time-series of temperature, winter precipitation sums, snowfall fraction, snowfall sums, mean and maximum snow depth starting in 2000 and ending between 2029 and 2099 for all case studies. See Table 1 for the acronyms of case studies.*

*Figure 9: Inter-annual variability for the REF (1980 to 2009), FUT1 (2010 to 2039), FUT2 (2040 to 2069) and FUT3 (2070 to 2099) periods expressed as the absolute standard deviation [cm] (left) and standard deviation relative to the mean [%] (right) for mean and maximum snow depth for the stations Adelboden (ABO), Engelberg (ENG), Davos (DAV), Weissfluhjoch (WFJ), Zermatt (ZER), Montana (MON), Scuol (SCU), Ulrichen (ULR). The boxes represent the inter-quartile range with the median as horizontal line. The whiskers represent 1.5 times the inter-quartile range and the dots represent outliers.*

4. - As a general recommendation, please avoid double parentheses (for example, line 69, or line 209, ...).
>>>> We removed the double parentheses:
*While it is important to estimate the contribution of ICV to uncertainties in future snow trends, it is however as important to investigate the inter-annual variability (IAV), of snow itself, which is defined as the year-to-year deviation from a long-term mean (He and Li, 2018) and which is likely to change under a future climate (IPCC, 2013).*

5. - Line 379-380: what do the percentages inside parentheses refer to? Scuol? If so, the sentence should be rephrased: "For Montana (Scuol), IAV increases from 50% (40%) ... up to more than 80% (70%)...".
>>>> We rephrased the sentence to: *For Montana IAV increases from 50% in the reference period up to more than 80% in the future 2 period. For Scuol we find an increase from 40% up to 70% between the two periods.*

6.   - Section 4.2: I understand that it is convenient to compare results of this study to another study (Schmucki et al., 2015) that focussed on the same stations. However, would it be possible to have some general comparison with other alpine studies outside of Switzerland?

>>>> We added a few studies to put our results into a global perspective. Adapted section 4.2:

[revised manuscript text omitted]

Technical corrections:
    7.   - Line 27: Please rephrase to: "In large parts of the world, water stored...".
>>>> We rephrased to: *In large parts of the world, water stored in snow…*

    8.   - Line 46: "Typically, studies compare different emission scenarios".
>>>> We rephrased to: *Typically, studies compare different emission scenarios…*

    9.   - Line 73: "the following hypotheses".
>>>> We corrected to *hypotheses*

    10. - Line 78: please remove the comma between "a" and "dynamically".
>>>> We removed the comma

    11. - Line 111: "the lack of".
>>>> We corrected to: *Due to the lack of measurements…*

    12. - Line 131: "The ClimEx LE consists of".
>>>> Corrected

    13. - Line 196: Please consider splitting this sentence into two after "parameters".
>>>> We rephrased the sentence to: *To evaluate the performance of the bias adjustment approach, we compared the statistical characteristics of observed meteorological variables to the bias-corrected ClimEx*

*LE for subdaily, daily, monthly and yearly values and compared the diurnal and annual regimes of the corrected parameters. We obtained good results in the calibration period 1984 to 2009.*

14. - Line 212-213: "from 1980 to 2009... from 2010 to 2039..., 2040 to 2069 ... and 2070 to 2099".
>>>> Corrected

15. - Line 266-267 (caption of Fig. 3): please replace the comma by a full stop.
>>>> We rephrased the sentence to: *Mean winter snowfall fraction based on observations and for the bias corrected ClimEx LE for the period 1984 to 2009. temp-th indicates the calibrated snowfall-rainfall separation threshold for the respective station. See Table 1 for the acronyms of case studies.*

16. - Line 274: please remove the comma after "clarify".
>>>> We removed the comma

17. - Line 291: please remove the comma after "Both".
>>>> We removed the comma

18. - Fig. 6: the title of the left column should read "Long-term winter mean".
>>>> We corrected the title to: *Long-term winter mean*

19. - Line 400: there seems to be a word missing before "just reach". Maybe "but"?
>>>> We corrected the caption to: *Figure 3: Mean winter snowfall fraction based on observations (obs; red) and for the bias-adjusted ClimEx LE (blue) for the period 1984 to 2009. temp-th indicates the calibrated snowfall-rainfall separation threshold for the respective station. See Table 1 for the acronyms of case studies.*

20. - Line 422: please remove the comma after "While".
>>>> We removed the comma

21. - Line 436: "we presented".
>>>> Rephrased to: *we presented*

22. - Line 446: please remove the comma after "found".
>>>> We removed the comma

23. - Line 446-447: please replace the two occurrences of the word "between" by the word "of".
>>>> We replaced "between" with "of"

24. - Line 470: please remove the comma after "station".
>>>> We removed the comma

25. - Line 472: "the present work".
>>>> Corrected to: *the present work*

26. - References: I noted a typo at line 676. Please check all references for other typos.
>>>> We removed the typo and checked all the other references and corrected where needed.

**Anonymous Referee #2:**

**27. General comments**

This paper presents the analysis of climate change trends of different variables related to the snow cover. This analysis includes data coming from a rare (if not unique) high resolution RCM large ensemble. It explores the inclusion of the Internal Climate Variability (ICV), available with such an ensemble, into the analysis of the trends of the variables and their significance. The paper also takes quick a look at the effect of climate change on the Inter Annual Variability (IAV). The analysis is done for 6 Swiss alpine stations and uses bias corrected RCMs fields used to drive the SNOWPAK snow model to obtain the snow-related variables.

I really enjoyed this paper. I thought it was very well done and I recommend its publication as I think it contributes to the general knowledge about the effect of climate change on snow.

>>>> We thank referee 2 for the very positive and helpful comments. We assiduously went through the comments and complemented and refined the manuscript accordingly.

**28. Specific Comments**

In the Introduction, the authors write about "downscaled model" but they do not specify that it is a Regional Climate Model (RCM). I think it should be corrected. In fact, RCM appears first on line 58 and is not defined.

>>>> We now introduce the term RCM in the introduction:
*Dynamically downscaled single model large ensembles, using a Regional Climate Model (RCM), are very rare. To our knowledge, only Fyfe et al. (2017) used snow water equivalent from a downscaled single model large ensemble to estimate the impact of ICV on near-term snowpack loss over the United States.*

29. On line 72 it is written that "Again, a single model large ensemble can help answer how inter-annual variability might respond to changes in climatic forcing". I do not disagree but a single model simulation could do the trick or an ensemble of one member simulations of different RCMs could also do the trick. This ensemble does not strike me as ideal for that task.

>>>> Thanks a lot for this comment. We certainly agree here, and we did not mean that our setup can fully answer such questions, but can rather – as said – HELP to answer such questions. Do clarify this point we changed the sentence to:
*Again, by complementing multi model-based approaches and by separating ICV from forced responses, a single model large ensemble can help answering how inter-annual variability might respond to changes in climatic forcing.*

30. On line 184 the authors write, "The same scaling factors are transiently applied to the distribution functions of 25-year slices of each ensemble member and pushed into the future by a ten year moving window." I am not sure I understand what it means.

>>>> We rephrased the sentence: *In a second step, we computed scaling factors between the simulated and observed data in the baseline period. For temperature, an additive scaling factor was estimated, while for the remaining variables multiplicative scaling factors were estimated. In a last step, for temperature the*

*scaling factors were added to the empirical distributions of each single ensemble member, while for the remaining variables the scaling factors were multiplied with the empirical distributions of the simulations. The same scaling factors were transiently applied to the distribution functions of 25-year slices of each ensemble member.*

31. On figure 2, is it me or the colours of the lines are inverted from left to right (i.e. blue on left is red on right)? I also wonder what the monthly mean figures add to the daily ones.

>>>> Thank you for the great observation skills. We changed the colors to the correct meanings.

32. For figure 7, I did not understand why the variable presented is snowfall fraction instead of snowfall. I understand the presentation of the validation of SNOWPAK is done with that variable but we have opposed trends for snowfall fraction and a signal for the total precipitation and we are interested into the mean and max snow depth. It seems to me it would be natural to look at total snowfall trends. I also think the sign of the trend should be visible on the figure or at least in the legend. For now we have to go look into the text to see if the trend is positive or negative.

>>>> Additionally to snowfall fraction we added snowfall [mm] in the analysis. Correspondingly we changed figures 7 and 8 (the sign of the trend is now also visible from the figures):

[Figure]

**Figure 1: Heat maps of trend significance for the Mann-Kendall test for different periods starting in 2000 and ending between 2029 and 2099, for winter mean temperature, winter precipitation sums, winter snowfall fraction, winter snowfall sums and winter mean and maximum snow depth, for the lowest station Engelberg (ENG) and the highest station Weissfluhjoch (WFJ).**

[Figure]

**Figure 2: Percentage of significant trends (p-value 5%) for time-series of temperature, winter precipitation sums, snowfall fraction, snowfall sums, mean and maximum snow depth starting in 2000 and ending between 2029 and 2099 for all case studies. See Table 1 for the acronyms of case studies.**

In the text we added: Sect. 2.6:

*As air temperature below a certain threshold occurring simultaneously with precipitation is the most important prerequisite of snowfall (Morán-Tejeda et al., 2013; Sospedra-Alfonso et al., 2015), we do not*

*only test for future snow depth trends, but also for the drivers of future snow conditions. For that reason, we apply the Mann-Kendall test also to temperature, precipitation, snowfall fraction and snowfall time-series.*

Sect. 3.4:

*Despite the strong mean climate change signal described in Sect. 3.3, section 3.4 points out the important role of ICV for the detection of statistically significant trends in future time-series of snow depth, and its most important drivers temperature, precipitation, snowfall fraction and snowfall. Note that, according to the poor validation results, we cannot draw conclusions on the absolute values of snowfall fraction, snowfall and snow depth for the stations Zermatt and Montana. Nevertheless, we can apply the trend test and compare relative changes. Figure 7 visualizes the results of the Mann-Kendall test for a positive trend in winter mean temperature and winter precipitation sums and a negative trend for winter snowfall fraction, snowfall, as well as winter mean and maximum snow depth for the lowest station Engelberg, and the highest station Weissfluhjoch for 1%, 5% and 10% significance levels.....*

*In contrast to snowfall fraction, we find a lower statistical significance for negative trends of total winter snowfall sums. For all stations, but Weissfluhjoch, over a period of 60 years, the percentage of statistically significant negative trends in winter snowfall sums is only between 25% and 60%. Over a period of 80 years between 60% and 90% of members show a statistically negative trend and by the end of the century only the stations Engelberg, Zermatt, Montana and Scuol show a trend significance for 100% of members. For the station Weissfluhjoch no statistically significant negative trend is obtained over any period. Even over 100 years less than 10% of members show a significant reduction in winter snowfall sums. These results imply that despite the strong temperature increase and a significant reduction of snowfall fraction, a reduction of total snowfall sums at this site remains very uncertain. ...*

*The stronger significance of mean snow depth compared to snowfall fraction and the higher significance compared to snowfall sums implies the combined effects of a very uncertain decrease of snowfall sums, combined with more rapid and more frequent snow melt.*

Sect. 4.2:

*We also find an uneven response of different snow metrics to anthropogenic warming. Statistically significant trends are first detected for mean winter snow depth, followed by winter snowfall fraction and later still by winter maximum snow depth; for trends in winter snowfall sums we can identify large uncertainties related to ICV. Our results are confirmed by Ishida et al. (2019) for three case studies in California, who investigate climate change impacts on interrelations between snow-related variables and find that temperature rise will affect, but will not dominate the future change in snow water equivalent and who also find uneven responses of different snow-related variables to anthropogenic forcing. Further, these results coincide with Pierce and Cayan (2013) and emphasize two points. First, also in the far future, we must expect considerable winter snowfall sums and events of large snow accumulation, even under RCP8.5. Overall a reduction of mean snow depth is rather driven by increased snowmelt than by a strong decrease in absolute snowfall sums. Consequently, trend detections for maximum snow depths over periods of less than 50 years largely depend on noise from ICV. Second, ICV remains the highest source of uncertainty over a short to medium range of time, but it can even hamper a statistically significant signal over periods of more than 50 years. On the other hand, with regard to future research, ICV cannot only reveal the possibilities of long hiatus periods, but it can also illustrate even faster snowpack declines in the Swiss Alps..*

33. For section 3.5, it was not clear to me if the mean used to compute the relative IAV is the mean of the reference period or the mean of every period. Generally speaking, this section was interesting but a bit light and unclear at the same time. I also think the relation between ICV and IAV could be better explored.

>>>> We made clear that the mean of each period was used to compute the relative IAV: *Nevertheless, in relative terms (relative to the mean of the corresponding periods), IAV can strongly increase in the future,...*

    34. On line 368 it is written "From Fig. 6 and Fig. 9 we can see that a gradual decrease of winter mean and maximum snow depth leads to a decrease in absolute IAV". What does this mean? I have problems with the "leads" word.

>>>> we rephrased the sentence to: *From Fig. 6 and Fig. 9 we can see that a gradual decrease of winter mean and maximum snow depth is accompanied by a decrease in absolute IAV.*

35. Line 414: "temporal variability was validated in Sect. 3.1". That is not clear to me, Sect. 3.1 presented the validation of snow depth time series for SNOWPAK (fig. 2), there is no question/measure of temporal variability, no?

>>>> It is correct, temporal variability after bias adjustment was validated in section 3.2, not 3.1.. We rephrased the sentence to : *Furthermore, QM cannot correct misrepresented temporal variability (Addor and Seibert, 2014). Therefore, inter-annual variability was validated in Sect. 3.2, yielding acceptable results.*

Technical corrections
I do not have any purely technical corrections to report.

---

## Referee Report (RR1)

On line 122 it is said that SNOWPAK does not use the precipitation data and on line 130 it is said that "A threshold of 50 % in relative humidity was set for all stations for rainfall/ snowfall discrimination". Does that mean that SNOWPAK sees rain or snowfall when the relative humidiy is over 50%?. I am confused. How does SNOWPAK deal with precipitation coming from the observation and the bias adjusted RCM data. Could the author clarify this?

---

## Editor Decision (ED1)

Dear authors

I am pleased with your revised manuscript. The referees have two small points I ask you to address. Some additional minor editorial points are listed below.

*Referee #1*

I only noted one technical detail that should be corrected before publication: the accronym for the Nash-Sutcliffe coefficiency is sometimes written "NSC" (e.g., line 214), sometimes "NSE" (e.g., line 247 and in Table 2). Please stick to one accronym throughout the manuscript.

*Referee #2*

On line 122 it is said that SNOWPACK does not use the precipitation data and on line 130 it is said that "A threshold of 50 % in relative humidity was set for all stations for rainfall/ snowfall discrimination". Does that mean that SNOWPAK sees rain or snowfall when the relative humidity is over 50 %? I am confused. How does SNOWPACK deal with precipitation coming from the observation and the bias adjusted RCM data. Could the author clarify this?

*Editorial comments*

Line 35 (and elsewhere in the manuscript): You may consider replacing "man-made".

Lines 74-78: The way the hypotheses and research questions are stated is a bit unusual and you never refer to the abbreviations (H1, etc) you introduce. You may as well just state: We hypothesize that ICV is a major … and IAV will change. Hence, we address the research questions: What are the uncertainties ... and how does…

Line 114: I am not sure whether you were actually considering snow transport by wind in your SNOWPACK simulation. If yes, I suggest you change "wind transportation" into "snow transport by wind". If no, I suggest you delete "wind transportation".

Line 248 (and elsewhere in the manuscript): My preference is elevation rather than altitude. "Altitude is for things that fly" I was once taught by a reviewer.

Line 531: You may consider adding "the high-elevation station" Weissfluhjoch. This would provide some explanation.

10 July 2020
Jürg Schweizer

---

## Author Response (AR2)

We thank all anonymous reviewers and the editor for their time and effort and the suggestions. The original comments of the reviewers are in blue. Our replies are in black.

**Anonymous Referee #1:**

I only noted one technical detail that should be corrected before publication: the accronym for the Nash-Sutcliffe coefficiency is sometimes written "NSC" (e.g., line 214), sometimes "NSE" (e.g., line 247 and in Table 2). Please stick to one accronym throughout the manuscript.

>>> corrected to NSE throughout the text

**Anonymous Referee #2**

On line 122 it is said that SNOWPACK does not use the precipitation data and on line 130 it is said that "A threshold of 50 % in relative humidity was set for all stations for rainfall/ snowfall discrimination". Does that mean that SNOWPAK sees rain or snowfall when the relative humidity is over 50 %? I am confused. How does SNOWPACK deal with precipitation coming from the observation and the bias adjusted RCM data. Could the author clarify this?

>>> we write in line 122 that we do not use the original measured precipitation data, but correct it for undercatch, before using it in SNOWPACK. The undercatch corrected precipitation data is then used as input for SNOWPACK. For bias adjustment we also used the undercatch corrected data as reference dataset.

In line 122 we write:

*As wind induced gauge undercatch underestimates precipitation, especially for mixed-, and solid precipitation, we do not use the original measured precipitation data to run the model. As described in Schmucki et al. (2014) precipitation was undercatch-corrected by applying a method developed by Hamon (1973), using a function of wind speed and temperature.*

**Editorial comments**

Line 35 (and elsewhere in the manuscript): You may consider replacing "man-made".

>>> we replaced it with "anthropogenic"

Lines 74-78: The way the hypotheses and research questions are stated is a bit unusual and you never refer to the abbreviations (H1, etc) you introduce. You may as well just state: We hypothesize that ICV is a major … and IAV will change. Hence, we address the research questions: What are the uncertainties ... and how does…

>>> we changed it accordingly

*We state the following hypotheses and aim at answering two research questions: First, ICV is a major source of uncertainty in trends of future Alpine snow depth. Our research question is: what are the uncertainties in future trends in Alpine snow depth attributed to ICV? Second, IAV of snow depth will change with anthropogenic climate forcing. Hence, our research question is: how does IAV of snow depth change with anthropogenic climate forcing?*

Line 114: I am not sure whether you were actually considering snow transport by wind in your SNOWPACK simulation. If yes, I suggest you change "wind transportation" into "snow transport by wind". If no, I suggest you delete "wind transportation".

>>> replaced it with *"snow transport by wind"*

Line 248 (and elsewhere in the manuscript): My preference is elevation rather than altitude. "Altitude is for things that fly" I was once taught by a reviewer.

>>> altitude was replaced with elevation

Line 531: You may consider adding "the high-elevation station" Weissfluhjoch. This would provide some explanation.

>>> we added "high-elevation station"